# Spatially Variant Error Elimination for High-Resolution UAV SAR with Extremely Small Incident Angle

Xintian Zhang [1], Shiyang Tang [1,*], Yi Ren [1], Jiahao Han [1], Chenghao Jiang [1], Juan Zhang [1], Yinan Li [2], Tong Jiang [2] and Qi Dong [3]

1    National Key Laboratory of Radar Signal Processing, Xidian University, Xi'an 710071, China; 21021110347@stu.xidian.edu.cn (X.Z.); 18021110250@stu.xidian.edu.cn (Y.R.); 20021211257@stu.xidian.edu.cn (J.H.); chjiang@stu.xidian.edu.cn (C.J.); jzhang@xidian.edu.cn (J.Z.)
2    China Academy of Space Technology (Xi'an), Xi'an 710100, China; liyn@cast504.com (Y.L.); jiangt@cast504.com (T.J.)
3    Beijing Institute of Control and Electronics Technology, Beijing 100038, China; dongqi456@gmail.com
*    Correspondence: sytang@xidian.edu.cn; Tel.: +86-029-8823-6159

**Abstract:** Airborne synthetic aperture radar (SAR) is susceptible to atmospheric disturbance and other factors that cause the position offset error of the antenna phase center and motion error. In close-range detection scenarios, the large elevation angle may make it impossible to directly observe areas near the underlying plane, resulting in observation blind spots. In cases where the illumination elevation angle is extremely large, the influence of range variant envelope error and phase modulations becomes more serious, and traditional two-step motion compensation (MOCO) methods may fail to provide accurate imaging. In addition, conventional phase gradient autofocus (PGA) algorithms suffer from reduced performance in scenes with few strong scattering points. To address these practical challenges, we propose an improved phase-weighted estimation PGA algorithm that analyzes the motion error of UAV SAR under a large elevation angle, providing a solution for high-order range variant motion error. Based on this algorithm, we introduce a combined focusing method that applies a threshold value for selection and optimization. Unlike traditional MOCO methods, our proposed method can more accurately compensate for spatially variant motion error in the case of scenes with few strong scattering points, indicating its wider applicability. The effectiveness of our proposed approach is verified by simulation and real data experimental results.

**Keywords:** unmanned aerial vehicle (UAV); synthetic aperture radar (SAR); spatially variant error; motion compensation (MOCO)

## 1. Introduction

Airborne synthetic aperture radar (SAR) is a valuable tool for remote sensing and mapping, providing high-resolution two-dimensional (2-D) images that improve detection performances [1–4]. Compared to traditional optical remote sensing, airborne SAR can be used for detection during the day, at night, and in adverse weather conditions, making it a flexible and reliable monitoring technology [5–7]. Recent advancements in unmanned aerial vehicle (UAV) technology have led to the development of micro-SAR devices that can be equipped on drones, offering advantages such as ease of operation and deployment and low cost, particularly in lightweight drones. UAV SAR can be used in hazardous conditions, such as during natural disasters or fires, to reduce the risks for rescue personnel [8–12].

A stable flying status is crucial for all kinds of airborne SAR systems to effectively synthesize the Doppler bandwidth. However, motion-induced error can compromise both resolution and overall system performance [13–16]. In practice, flight paths are often non-linear due to atmospheric airflow, resulting in motion error that significantly impacts the Doppler characteristics of the echo data, including the Doppler centroid and the Doppler chirp rate, which determine the azimuth position and depth of field, respectively. These

factors are inherently limited by residual range cell migration (RCM) and nonlinear phase error (NPE) of the target [17–20]. As a result, motion error cannot be neglected during airborne SAR imaging processing.

For stable aircraft, such as transport planes, the impact of motion error on SAR performance is generally negligible [21–23]. However, for drones, atmosphere turbulence can cause a significant motion-induced error compared to manned aircraft [24]. Moreover, due to payload limitations, drone SARs are typically designed with higher frequencies to shorten the wavelength and reduce the size and weight of the RF device, resulting in significant phase error caused by motion error [25]. In addition, compared to traditional airborne SAR systems, drone SAR systems have a shorter detection range, requiring a larger antenna pitch angle to cover the ground scene. This results in significant spatial variability caused by motion error, which can greatly degrade imaging quality, particularly in high-resolution applications.

With the continuous improvement of radar resolution, the demand for enhanced accuracy in SAR motion compensation has grown [26]. Currently, the measurement accuracy of inertial guidance systems (INS) or global positioning systems (GPS) [27] often cannot meet the requirements for high-resolution SAR motion compensation UAV systems. This limitation restricts the use of motion compensation algorithms based on navigation information. Consequently, motion compensation emerges as a critical factor in obtaining high-resolution images for UAV SAR systems. Previous research has primarily focused on error properties and estimation methods, such as the phase gradient autofocus (PGA) technique for spotlight mode [28] and motion compensation methods for strip mode [29]. However, these approaches have limitations in addressing scenarios with extremely small angles of incidence. Therefore, the utilization of autofocus approaches is recommended to implement motion compensation in drone SAR systems, particularly for close-range scenarios with extremely small angles of incidence. Further research is necessary to investigate high-resolution imaging and motion compensation algorithms tailored for such scenarios.

Various studies have investigated the SAR autofocus problem, with phase gradient autofocus (PGA) being one of the most well-known techniques [30,31]. Qualification PGA (QPGA) reduces the requirement for the number of salient points in two dimensions, and increasing the number of salient points can improve the precision of phase gradient estimation and the robustness of the algorithm [32]. Different weighting strategies have been proposed to address the issue of low signal clutter ratio (SCR) features in phase gradient estimation and enhance the contribution of high-quality features. However, most existing algorithms have been proposed to compensate for the spatial invariant motion error and do not address the issue of spatial variation in the scene, which may limit their practical performance. In the case of large elevation angles, the observation distance to the target is small, i.e., the slant range is smaller than in the case of small elevation angles. Indeed, the impact on the echo signals is also significant. Therefore, there is a pressing need for a more precise and accurate motion compensation method. Moreover, due to the presence of spatial variation, the variation in slant range in the range dimension is smaller in the case of large elevation angles than in the case of small elevation angles. This leads to increased range spatial variation in the echo, while imaging targets are located at different range cells in the scene. As a consequence, existing algorithms that only consider spatial variation fail to address this issue adequately. Hence, the development of a more accurate higher-order autofocusing algorithm is necessary.

This study proposes a MOCO algorithm for UAV SAR systems that addresses practical issues, including range motion error and PGA failure. It first establishes a geometric model of the system and analyzes the potential issues related to motion error and inadequate scattering points. Then, it proposes a motion compensation algorithm based on an improved phase-weighted estimation PGA algorithm, which is able to estimate both spatial invariant and spatially variant phase error and perform full aperture phase stitching. Finally, a combined autofocus method is proposed to address the issue of insufficient strong scattering points in the scene, which selects different autofocus methods based on the proportion of

strong scattering points and sets a threshold to improve the spatially variant performance of MOCO. Experimental results show that this proposed method has a wider application and a higher imaging precision compared with traditional methods.

In summary, the innovation and contribution of this work is a MOCO strategy designed for UAV SAR high-resolution imaging in extremely small incident angle. The core of which is the statistical threshold selection, resulting in a combined autofocus method applicable to arbitrary imaging scene. By selecting the appropriate processing method, the proposed approach addresses the challenge of poor performance of traditional methods when there are few strong scattering points in the imaging scene. Meanwhile, to ensure the accuracy of MOCO when incident angle is extremely small, an improved PGA algorithm that considers the effect of the high-order phase errors is utilized, which further enhance the robustness and effectiveness of the proposed approach.

The rest of this paper is organized as follows: Section 2 analyzes the airborne SAR motion error both geometrically and mathematically. In Section 3, an improved combined MOCO approach based on spatial variation, consisting of three parts (i.e., a statistical threshold selection, an improved phase-weighted estimation PGA algorithm, and an auxiliary algorithm), is presented in detail. Section 4 provides the experimental results including the simulation and real data processing, and Section 5 presents the conclusion summarizing the main findings.

## 2. Modeling

### 2.1. Geometric Model

The geometric model of airborne SAR imaging with motion error caused by atmosphere turbulence is shown in Figure 1.

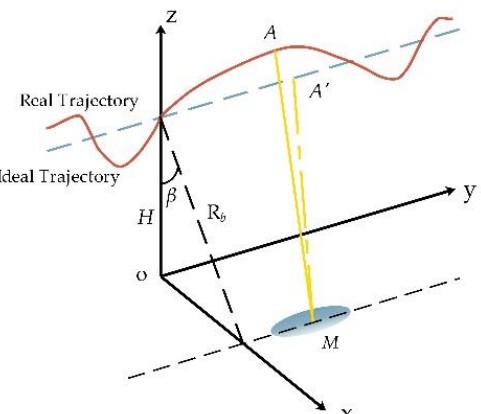

**Figure 1.** SAR geometric model with motion error.

In a spatial coordinate system based on the right-handed convention, the origin $O$ is set as the ground projection of the synthetic aperture central. The platform is assumed to fly along $y$-axis at a constant velocity $v$ and altitude $H$. The area of interest on the ground lies in the right field of view of the aircraft. $M$ denotes the arbitrary point target in the imaging scene, $R_b$ denotes the closest range of $M$, and $\beta$ is the corresponding incident angle, i.e., $\beta = \arccos(H/R_b)$.

It is almost impossible for an aircraft to maintain a constant attitude. Unstable motion leads to uncertain error, causing the real air path to deviate from the expected path as shown by the red solid line and blue dashed line plotted in Figure 1, respectively. $A$ and $A'$ are the arbitrary point of the platform on the real path and the expected one, respectively. Thus, the instantaneous position on the real and the expected paths can be expressed by the spatial coordinates $x$, $y$, and $z$, i.e., $[x(\eta), y(\eta), z(\eta)]$ and $[0, v\eta, H]$, where $\eta$ denotes the slow time and $v$ denotes the imaging velocity. Additionally, the location of $M$ can

be defined as $[x_n, y_n, z_n]$. Thus, the instantaneous range from $A$ to $M$ can be expressed as follows:

$$
\begin{aligned}
R(\eta, R_b) &= \sqrt{[x(\eta) - x_n]^2 + [y(\eta) - y_n]^2 + [z(\eta) - z_n]^2} \\
&= \sqrt{[x(\eta) - x_n]^2 + y(\eta)^2 + z(\eta)^2 + R_b^2 - 2y(\eta)R_b \sin\beta - 2z(\eta)R_b \cos\beta} \\
&\approx \sqrt{[x(\eta) - x_n]^2 + R_b^2 - 2y(\eta)R_b \sin\beta - 2z(\eta)R_b \cos\beta}
\end{aligned} \tag{1}
$$

where $\eta$ is azimuth slow time, $R_b = \sqrt{y_n^2 + z_n^2}$, and $y_n = R_b \sin\beta$, $z_n = R_b \cos\beta$. Meanwhile, Equation (1) can be expanded by the Taylor series as

$$
R(\eta) = R_b + \frac{[x(\eta) - x_n]^2}{2R_b} - y(\eta)\sin\beta - z(\eta)\cos\beta \tag{2}
$$

Thus, $R(\eta, R_b)$, corresponding to azimuth and range dimensions, is decomposed into three components, which are explained in detail as follows:

- The first component $R_b$ denotes the slant range of $M$ at aperture central and determines the range position of $M$ on the SAR image;
- Note that the second component with respect to $R_b$ mainly depends on the $x$-axis motion status, and varies with the azimuth position of $M$. This term determines the azimuthal position of $M$ on the SAR image. In reality, motion error on the $x$-axis may deteriorate the linear Doppler central and further cause a shift in the result. In addition, apart from phase error, the difference interval between chirps caused by motion error on the $x$-axis will result in azimuthal non-uniformity;
- The third component affected by $y(\eta)$ and $z(\eta)$ is called the cross-path error, which is an important component that needs to be compensated primarily during imaging processing. This is because the speeds along the $y$-axis and $z$-axis change as the platform approaches the target, thereby deteriorating the Doppler frequency, such as the Doppler central and the Doppler chirp rate. Thus, the result is shifted and defocused. Moreover, it should be noted that the component $\beta$ reflects the spatially variant nature, leading to extra range cell migrations (RCM) and non-linear phase errors (NPE). As a result, this component will cause a great impact on processing.

By means of Equation (2), the range history error $\Delta R(\eta)$ between the real and the expected trajectory can be expressed as

$$
\begin{aligned}
\Delta R(\eta) &= R(\eta) - R'(\eta) \\
&= R_b + \frac{[x(\eta) - x_n]^2}{2R_b} - y(\eta)\sin\beta - z(\eta)\cos\beta - \left[R_b + \frac{(v\eta - x_n)^2}{2R_b}\right] \\
&= \frac{[2v\eta - 2x_n + \Delta x(\eta)]\Delta x(\eta)}{2R_b} - y(\eta)\sin\beta - z(\eta)\cos\beta
\end{aligned} \tag{3}
$$

where $\Delta x(\eta)$ denotes the position error along the $x$-axis at any moment.

Figure 2 shows the range error corresponding to range and azimuth directions caused by $\Delta x(\eta)$ in a typical UAV SAR application, with different reference ranges $R_s$, where the speed along the $x$-axis varies from $-1$ m/s to $1$ m/s during the entire aperture. It is apparent that the range error denoted by $\Delta x(\eta)$ is too small to cause defocusing in the images even in high-resolution cases. Thus, these errors can be ignored.

Hence, Equation (3) can be simplified as

$$
\Delta R(\eta) = -y(\eta)\sin\beta - z(\eta)\cos\beta \tag{4}
$$

However, $\Delta R(\eta)$ is not only determined by $y$-axis and $z$-axis motion error but depends on $\beta$ as well. This implies that $\Delta R(\eta)$ is cross-coupled and spatially variant in Equation (4).

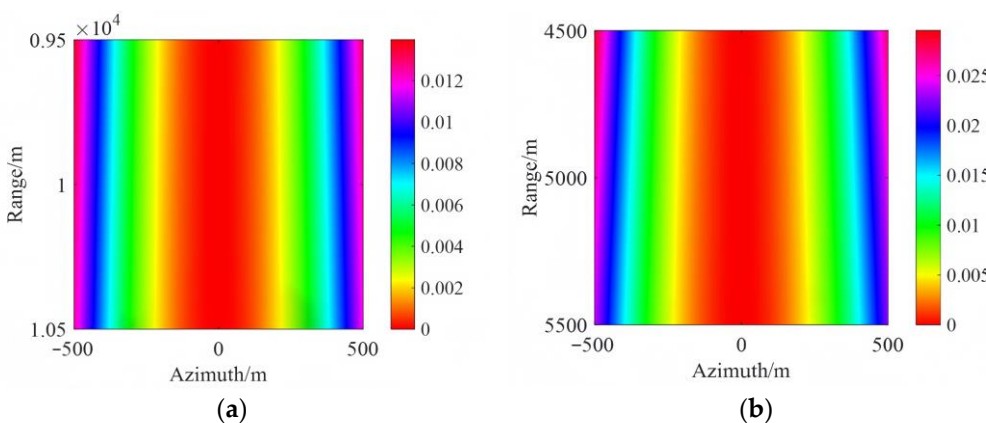

**Figure 2.** The range error caused by the *x*-axis motion error. (**a**) $R_s$ = 10 km. (**b**) $R_s$ = 5 km.

### 2.2. Spatially Variant Error Analysis

Before developing motion compensation algorithms, it is essential to analyze the spatial variation error that results from motion error. In UAV applications, these errors exhibit more pronounced characteristics compared to traditional airborne SAR imaging. This disparity can be attributed to the large antenna pitch required to cover the ground scene, which is limited by the detection power. Figure 3 shows the relationship between antenna pitch and slant range. Generally, when the phase error resulting from spatially variant error is larger than $\pi/4$, it has a noticeable impact on the imaging process. We refer to this distance as the "near" detection areas. Conversely, when the phase error within the scene is less than $\pi/4$, the impact on imaging can be disregarded, and this distance is termed the "far" detection area.

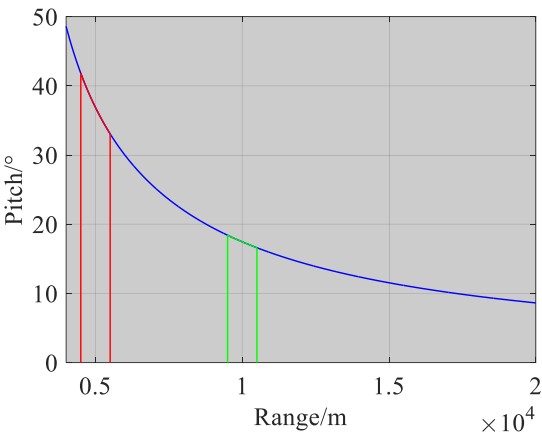

**Figure 3.** The relationship of antenna pitch with respect to the slant range.

In Figure 3, the red and green labels indicate near and far detection areas, respectively. Although the amplitudes of the red and green detections areas are the same, it is evident that the rate of change in the red detections areas is significantly greater than that in the green detections areas under a large elevation angle. This signifies that the phase error induced by spatially variant error in the far detections areas can be disregarded.

To simplify the analysis, let us assume that $r_s$ denotes the reference range vector from the platform to the interested area central at antenna phase central (APC), $r$ denotes the range vector from the platform to $M$, which can be expressed as

$$|r| = |r_s + \Delta r| \tag{5}$$

where $\Delta r$ is the range vector from $M$ to the scene center, which can be defined as the spatially variant part of $M$ relative to the center, i.e., $|\Delta r|$.

To further analyze the spatial variation induced by motion error, $\sin\beta$ and $\cos\beta$ in Equation (6) can be expanded by the Taylor series as

$$
\begin{aligned}
\cos\beta &= \frac{H}{|r_s + \Delta r|} \\
&= \frac{H}{|r_s|} - \frac{H}{|r_s|^2}|\Delta r| + \frac{H}{|r_s|^3}|\Delta r|^2 - \frac{H}{|r_s|^4}|\Delta r|^3 + \frac{H}{|r_s|^5}|\Delta r|^4 \cdots \\
\sin\beta &= \sqrt{1 - \cos^2\beta} \\
&= 1 - \frac{H^2}{2|r_s|^2} - \frac{H^4}{8|r_s|^4} + \left(\frac{H^2}{|r_s|^3} + \frac{H^4}{2|r_s|^5}\right)|\Delta r| - \left(\frac{3H^2}{2|r_s|^4} + \frac{5H^4}{4|r_s|^6}\right)|\Delta r|^2 \\
&\quad + \left(\frac{2H^2}{|r_s|^5} + \frac{5H^4}{2|r_s|^7}\right)|\Delta r|^3 - \left(\frac{5H^2}{2|r_s|^6} + \frac{35H^4}{8|r_s|^8}\right)|\Delta r|^4 \cdots
\end{aligned}
\tag{6}
$$

Figure 4a,b display the spatially variant error corresponding to the azimuth and range directions for different range cases based on Equation (6), where the speed along the $x$-axis and $z$-axis varies from $-1$ m/s to 1 m/s during the whole aperture. It is clear that the spatially variant error increases as the reference range decreases.

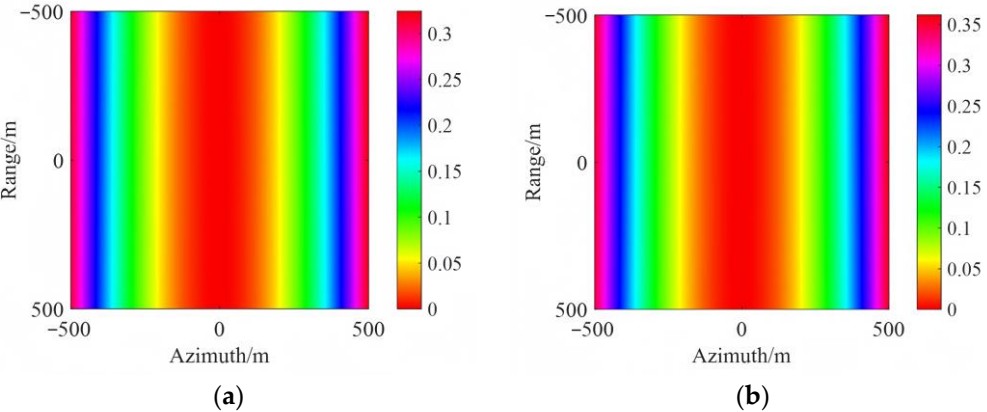

**Figure 4.** Spatially variant error: (**a**) $R_s$ = 10 km. (**b**) $R_s$ = 5 km.

The effects of spatial variation caused by motion error in UAV SAR are more pronounced than in conventional airborne SAR platforms, particularly in high-resolution applications, and thus cannot be overlooked.

### 2.3. Discussion of PGA Performance

Autofocus techniques are essential for improving the depth of field in practical airborne SAR processing. The reason is that, due to the presence of system noise, NPE caused by motion error cannot be fully compensated for by motion sensors. Additionally, high-frequency errors, which are associated with fine-scale variation in the motion trajectory, cannot be accurately measured by INS or GPS. Among the nonparametric autofocus methods, PGA has gained widespread usage in most airborne SAR systems due to its excellent performance. The critical step in PGA is to obtain NPE from the phase error gradient.

For further analysis, let us assume the $n$-th range cell's data after windowing and shifting is denoted by $g_n(\eta)$, the one after inverse Fourier transform (IFT) is given by $G_n(f_\eta)$, and the scatter-dependent phase function is denoted by $\theta_n(f_\eta)$. Thus, the linear unbiased minimum variance (LUMV) estimation of the phase error gradient $\phi(f_\eta)$ is given by [27]:

$$
\begin{aligned}
\phi(f_\eta) &= \frac{\sum_n \mathrm{Im}\left[G_n^*(f_\eta)\dot{G}_n(f_\eta)\right]}{\sum_n |G_n(f_\eta)|^2} \\
&= \dot{\phi}(f_\eta) + \frac{\sum_n \left[|G_n(f_\eta)|^2 \dot{\theta}_n(f_\eta)\right]}{\sum_n |G_n(f_\eta)|^2}
\end{aligned}
\tag{7}
$$

where $\sum_n(\cdot)$ denotes the summation operation, $\dot{G}_n(f_\eta)$ denotes the first-order derivative of $G_n(f_\eta)$, $G_n^*(f_\eta)$ denotes the conjugate of $G_n(f_\eta)$, $\dot{\phi}(f_\eta)$ denotes the first-order derivative of $\phi(f_\eta)$, and $\dot{\theta}_n(f_\eta)$ denotes the first-order derivative of $\theta_n(f_\eta)$. Based on Equation (7), NPE $\theta(\eta)$ can approached the true value through iterative correction. However, there are two weaknesses:

According to Equation (4), PGA is a discrete point-type autofocus algorithm that averages over a number of samples (i.e., range cells) and neglects range-independent error during processing. However, as previously analyzed, the range variant error becomes more prominent with a larger antenna pitch compared to a smaller pitch. Therefore, the traditional PGA may not be suitable for such scenarios.

Moreover, classic PGA relies on selecting strong range cells as samples to ensure accuracy, which implies that the ability to focus an image depends entirely on the absence of dominant reflectors. However, this approach lacks robustness, especially for featureless areas. As depicted in Figure 5, the area within the yellow dash box represents a scene with few strong scattering points. In most drone SAR applications, obtaining enough features from the small interested area is impossible. Therefore, the performance of PGA should be improved.

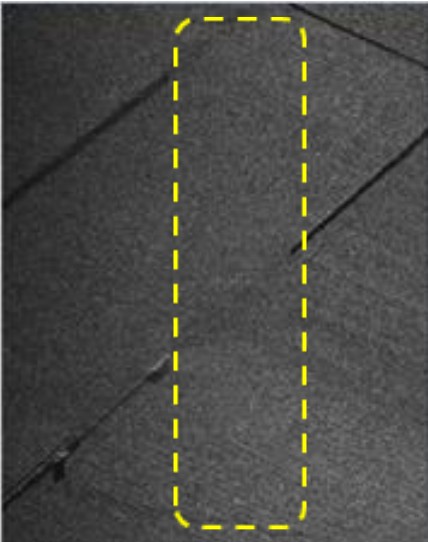

**Figure 5.** Focusing quality of the classic PGA for featureless.

## 3. Approach

In this section, we propose a combined autofocus approach that leverages statistical techniques to address the limitations of existing autofocus methods. Our proposed approach utilizes a brightness-counting method to obtain clearer and higher-quality images. To further enhance the stability and quality of the algorithm, we introduce a statistical threshold.

### 3.1. Statistical Threshold Selection

As previously discussed, the accurate selection of strong scattering points is crucial for the successful implementation of PGA. However, in practice, weak or non-existent scattering points are often selected, which may result in the inability to properly determine suitable scattering points and reduce image accuracy. Thus, it is imperative to improve the performance of selecting strong scattering points to enhance the quality of images.

The brightness of SAR images is typically represented by the intensity of the scattering points following mean quantization. To accurately quantify the intensity of each point, we developed a statistical threshold value that utilizes the statistical histogram approach. Specifically, the image brightness is partitioned into a range of 0 to the maximum value $h$,

determined based on the specific situation. Then, we analyzed the intensity distribution of points in the image and derived a probability density function of the intensity distribution.

To identify the differences in probability density functions of target strength distribution for different scenarios, statistical analysis was performed on the target strength distribution of various scenes.

Assuming a SAR image has $m$ azimuth samples and $n$ range samples, the intensity value corresponding to each pixel point of the image is first computed and quantized to its mean value. In this paper, the gray interval $h$ is set to 255, and the intensity of a pixel point in the $k$-th interval is denoted as $x_k$, $(0 \leq k \leq 255)$. Therefore:

$$x_0 + x_1 + \cdots + x_{255} = mn \tag{8}$$

Then, the probability density function of the intensity distribution for each point in the image is denoted as $f(k)$:

$$f(k) = \frac{x_k}{mn}, (0 \leq k \leq 255) \tag{9}$$

Additionally, the corresponding threshold of the image can be calculated from the obtained probability density function, i.e.:

$$\text{T} = \sum_{0}^{k_0} f(k) \tag{10}$$

where $k_0$ can be regarded as the image intensity demarcation range, which is determined based on the specific circumstances encountered in the SAR image analysis. From Equation (10), it indicates an abundance of strong scatterers in the image under the condition that the proportion is smaller than a value $p$, which necessitates compensation by employing an improved phase-weighted estimation PGA algorithm. Conversely, if the proportion is larger than or equal to $p$, further compensation is necessary by employing an auxiliary algorithm.

Unlike the traditional algorithm, the imaging method with the threshold selection addresses the challenge of poor performance of the traditional PGA algorithm, which heavily relies on the selection of strong scatterers in the image. This threshold-based imaging method offers improved performance and can be applied to a wider range of scenarios.

In this method, the imaging process is enhanced by considering the count of all scattering points in the scene and analyzing the brightness distribution. From the above analysis, the proportion of strong scatterers in the image can be obtained. By comparing this proportion with the threshold value, different compensation strategies are selected for achieving high-resolution imaging. In practice, the imaging method improves the imaging performance of the traditional autofocusing algorithm and is verified in the real-data experiment in Section 4.

### 3.2. Improved Phase-Weighted Estimation PGA

After analyzing the motion error occurring at large elevation angles, it becomes apparent that the phase error, resulting from the spatial variation due to the motion error, cannot be ignored. The conventional PGA algorithm, which assumes that motion error do not vary spatially with range, is unsuitable for imaging blind spots at large elevation angles. In this section, we propose an improved phase-weighted estimation PGA algorithm and a novel MOCO method based on this algorithm. To solve the spatially variant error, we selected range cells with high contrast from the background.

To acquire the phase error, a spatially variant matrix is first constructed using least squares estimation. At large elevation angles, it is necessary to analyze the impact of the spatial variation of the high-order term range. By substituting Equation (6) into Equation (4), we establish a relationship between motion error and range, encompassing terms ranging from the first-order to the fourth-order spatial variation. The data simulation presented in Table 1 yields Figure 6, which illustrates the effect of the phase error from the

spatial variation in the range up to the fourth order. The contour plot in Figure 6 employs units of $\pi$. In general, the motion errors can be ignored when they are less than $\pi/4$. From the simulation results, it can be seen that the phase error of the first-order to the third-order range spatial variation significantly exceeds $\pi/4$, while the phase error of the fourth-order range spatial variation is considerably less than $\pi/4$. In other words, the phase error from the first-order to the third-order range spatial variation should not be considered negligible.

**Table 1.** Simulation Parameters.

| Parameters | Value |
|---|---|
| Carrier Frequency | 35 GHz |
| Pulse Repeat Frequency | 625 HZ |
| Bandwidth | 1200 MHz |
| Pulse Width | 0.54 μs |
| Sampling Frequency | 1440 MHz |
| Reference Slant Range | 4000 m |
| Height | 3000 m |
| Squint Angle | 0 rad |
| Velocity | 40 m/s |

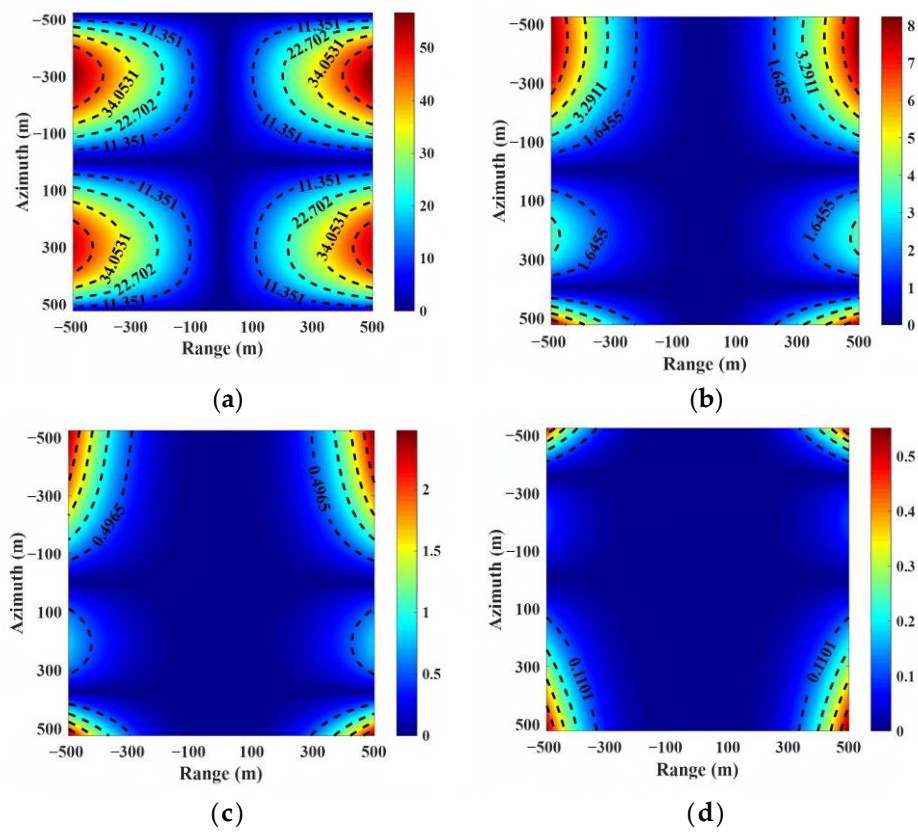

**Figure 6.** Phase error from first-order to fourth-order range spatial variation: (**a**) first-order; (**b**) second-order; (**c**) third-order; (**d**) fourth-order.

Consequently, the phase error can be modeled as a third-order polynomial with respect to range, i.e.:

$$\Phi(\eta, R_b) = b_0(\eta) + b_1(\eta)\Delta r + b_2(\eta)\Delta r^2 + b_3(\eta)\Delta r^3 \tag{11}$$

where $\eta$ represents the azimuth slow time and $\Delta r$ represents the difference between the slant range of an arbitrary target in the scene and the slant range of the scene center. $b_0$, $b_1$, $b_2$, and $b_3$ represent the constant, first-order, second-order, and third-order coefficients of

phase error, respectively. The phase error function is spatially variant in range due to the presence of motion error.

Once the expression of phase error is obtained, a weighted least squares estimation of $b_0(\eta)$, $b_1(\eta)$, $b_2(\eta)$, and $b_3(\eta)$ can be formulated as:

$$
\begin{aligned}
\hat{B} &= \left(A^T M A\right)^{-1} A^T M \Phi \\
&= \begin{bmatrix} \hat{b}_0(\cdot) \\ \hat{b}_1(\cdot) \\ \hat{b}_2(\cdot) \\ \hat{b}_3(\cdot) \end{bmatrix}_{4 \times L}
\end{aligned}
\tag{12}
$$

where, $\hat{b}_0(\cdot)$, $\hat{b}_1(\cdot)$, $\hat{b}_2(\cdot)$, and $\hat{b}_3(\cdot)$ are the gradient estimation of $b_0(\eta)$, $b_1(\eta)$, $b_2(\eta)$, and $b_3(\eta)$, respectively, $L$ represents the azimuth length of the sample, and $M = diag[m_1, m_2, \cdots, m_k]$ denotes the contrast weighting matrix between the target and background, where $m_k$ is the contrast within the $k$-th range cell. $A$ denotes the matrix of the spatially variant range, which can be expressed as:

$$
A = \begin{bmatrix} 1 & \Delta r(1) & \Delta r(1)^2 & \Delta r(1)^3 \\ \vdots & \vdots & \vdots & \vdots \\ 1 & \Delta r(K) & \Delta r(K)^2 & \Delta r(K)^3 \end{bmatrix}_{K \times 4}
\tag{13}
$$

The phase gradient estimation matrix of the selected sample is expressed as:

$$
\Phi = \begin{bmatrix} \hat{\phi}(1, \cdot) \\ \vdots \\ \hat{\phi}(k, \cdot) \end{bmatrix}_{K \times L}
\tag{14}
$$

where $\hat{\phi}(k, \cdot)$ is the phase gradient estimation of the $k$-th sample cell.

The entire phase gradient estimation algorithm above utilizes the spatial invariant phase error in the range sub-apertures and estimates the phase gradient in each sub-aperture with high precision through maximum likelihood estimation (MLE). The polynomial coefficients can then be estimated using least squares estimation, thereby realizing the estimation of the spatially variant phase error of the range. As proposed in the improved PGA algorithm, the corresponding MOCO method for high-resolution SAR can effectively achieve both phase and envelope compensation while accurately compensating for spatially variant error.

It is recommended to use the overlapped sub-apertures strategy in real data processing to enhance the accuracy and robustness of the improved phase-weighted estimation PGA algorithm. The improved PGA algorithm can achieve highly accurate phase error estimation with high operational efficiency and robustness.

### 3.3. Auxiliary Algorithm

In this subsection, an auxiliary algorithm based on the quadratic phase error model [33,34] is designed to address the issue of the PGA algorithm failing in cases where there are few features available for accurate phase error estimation. This auxiliary algorithm can effectively compensate for such a deficiency with high estimation accuracy.

Unlike the PGA algorithm, the auxiliary algorithm is independent of the number of features. It is based on a parameter model and can effectively estimate the Doppler modulation frequency while accurately compensating for quadratic phase error. Moreover, it can produce better focusing effects for scattering points when the motion error is small. The auxiliary algorithm is characterized by its ability to achieve high-quality results with low computational complexity, as it only requires operations such as FFT, IFFT, correlation, and complex multiplication [35–37]. Additionally, the algorithm does not require multiple

iterations and can adapt to various imaging scenarios easily. Concomitantly, the proposed method exhibits robustness in estimating the Doppler velocity parameter.

Assuming that the residual phase error remains spatially invariant, the algorithm is employed to extract instantaneous Doppler velocities within a sub-aperture, followed by double integration of the Doppler velocities to obtain the aperture's complete phase function. The specific algorithm is described as follows:

The spatial invariant phase error obtained by estimation is $\phi_{ne}$, the spatial invariant range migration is $\Delta R(\eta) \approx \phi_{ne}\lambda/4\pi$, and the phase function of coarse compensation is:

$$H(\eta) = \exp\left( j4\pi(f_r + f_c)\frac{\Delta R(\eta)}{c} \right) \tag{15}$$

The change in azimuth modulation frequency can be obtained using the auxiliary algorithm. It can be deduced that the second-order integration of the frequency modulation with respect to time yields the phase. Thus, the phase error can be obtained by integrating this variation $\Delta K_a$. Note that the first-order integral of the frequency modulation yields the frequency while removing its linear component, which helps prevent linear offset of the image orientation. The phase error $\phi_{ne}$ can then be expressed as:

$$\phi_{ne}(n) = 2\pi\sum_{n=0}^{N} \Delta f_a(n) \cdot \Delta T_a^2 = 2\pi\sum_{n=0}^{N}\sum_{n=0}^{N} \Delta K_a(n) \cdot \Delta T_a^2 \tag{16}$$

where $N$ represents the number of points in the azimuth direction, and $\Delta f_a$ represents the frequency shift. Through this method, the spatial invariant phase error can be accurately compensated even if the inertial navigation system (INS) fails.

### 3.4. Flowchart of Imaging Approach

For complex scenes, the conventional autofocus algorithm based on the image criterion may not adequately account for the spatial variability of motion error and may fail to meet focus requirements. To address these issues, a threshold-based combined imaging algorithm for high-resolution SAR is proposed.

Drawing upon the derivation in the preceding section, a statistical threshold is selected to design a composite autofocus algorithm with a broader scope of application. This approach enables the use of the PGA algorithm in scenes where the image's brightness features exceed a certain threshold. In contrast, for scenes where distinct features are not present and the image's brightness falls below a designated threshold, such as deserts and grasslands, an auxiliary algorithm can be utilized for imaging.

The flow chart of the combined autofocus algorithm proposed in this paper is shown in Figure 7. This combined focusing approach ensures both sufficient focusing accuracy in scenes with an abundance of strong points and stable focusing in scenes with sparse scatterers. Additionally, this approach has a wider range of applications and higher stability.

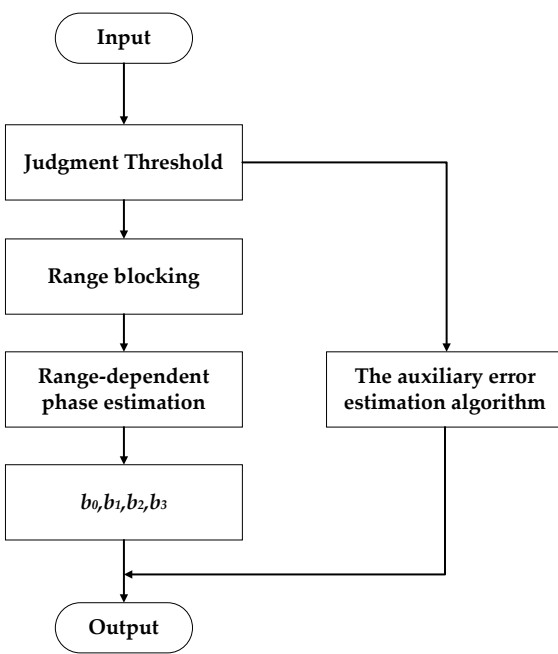

**Figure 7.** Flow chart of combined autofocusing algorithm.

## 4. Experiment

In this section, the simulation and real-data processing results are presented to demonstrate the performance of the proposed approach.

### 4.1. Simulation Results

In this subsection, a simulation experiment with point targets is conducted to evaluate the performance of the proposed algorithm. An area with dimensions of $500 \times 500$ m is selected for the experiment, as shown in Figure 8. Two point targets are placed at the center and edge of the imaging scene, respectively. The simulation parameters are shown in Table 1 and the platform is flying along the *y*-axis.

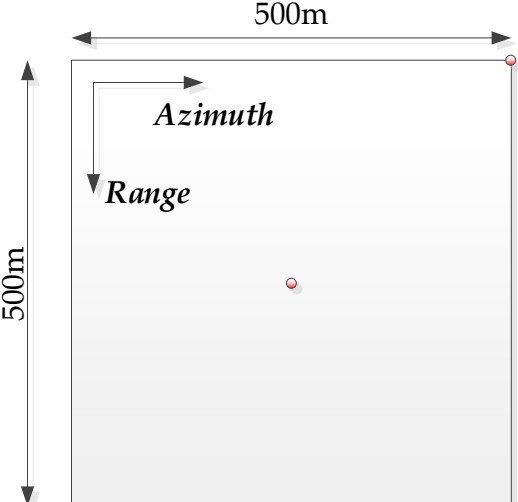

**Figure 8.** Simulation scene.

Figure 9a–c illustrates the three-dimensional instantaneous velocity parameters of the UAV obtained through simulations, with varying orders of acceleration. The results demonstrate that the motion state of the platform is unstable, and thus the motion error



cannot be neglected. However, due to the large antenna pitch, the spatial variation is significant, as evidenced by Figure 10. Therefore, spatial variability cannot be overlooked.

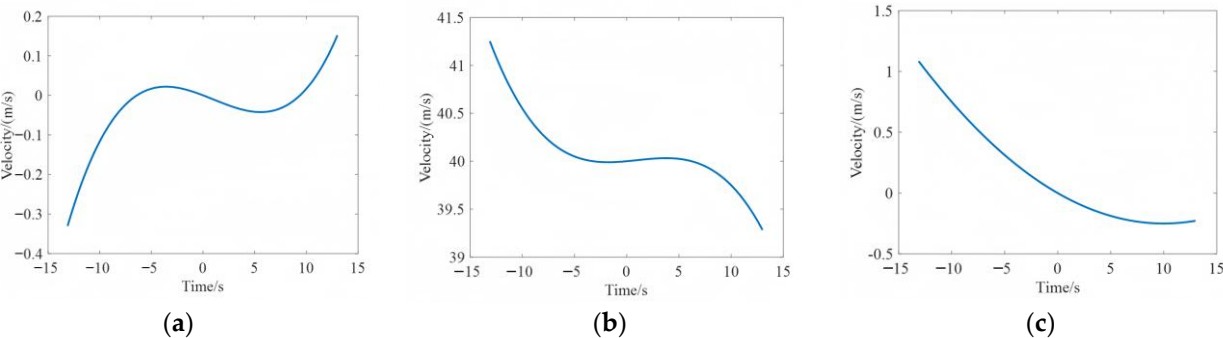

**Figure 9.** Simulated velocities along three axes: (**a**) *x*-axis; (**b**) *y*-axis; (**c**) *z*-axis.

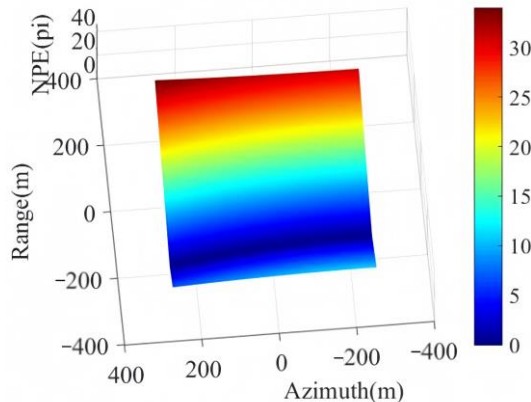

**Figure 10.** Spatially variant error caused by motion error.

In the experiment, the image intensity demarcation range $k_o$ is set to 150, and the value $p$ is set to 0.9. The probability function $f(k)$ is then integrated between 0 and 150. For probability density functions greater than 0.9, it can be inferred that all point targets in the scene are sufficiently weak. The integration method can provide a theoretical foundation for determining the threshold value and designing the subsequent combined autofocus algorithm.

Figure 9 displays the velocity error added in three directions, indicating that the velocity changes are relatively uneven. In Figure 11, direct imaging results obtained using the conventional algorithm are presented. The simulation graph shows that the central point target achieves a relatively good focusing effect, while there is some degree of defocusing at the edge points. In Figure 12, imaging results using the improved algorithm are shown. Figure 12a displays the focusing performance of the central point, and Figure 12b displays that of the edge points. The azimuth imaging quality values with motion error are listed in Table 2.

Comparing the focusing results in Figures 11 and 12 and the imaging data of the azimuth focusing parameters in Table 2, we can find that both the conventional method and the proposed method have great focusing effects on the central point target. However, it is evident from Figure 11b that the central point target achieves a relatively good focusing effect, while there is some degree of defocusing at the edge points. Additionally, the PLSR and ISLR indices of the conventional method deviate significantly from the ideal values, which can be attributed to ignoring the effect of high-order spatially variant motion error. In contrast, after applying our proposed method, as illustrated in Figure 12b, the edge points are well-focused, and the PLSR and ISLR indicators are basically consistent with the ideal values. After compensation, all point targets are well focused. The simulation results

indicate that both the central and edge points meet the focusing requirements, verifying the effectiveness and advantages of the proposed method.

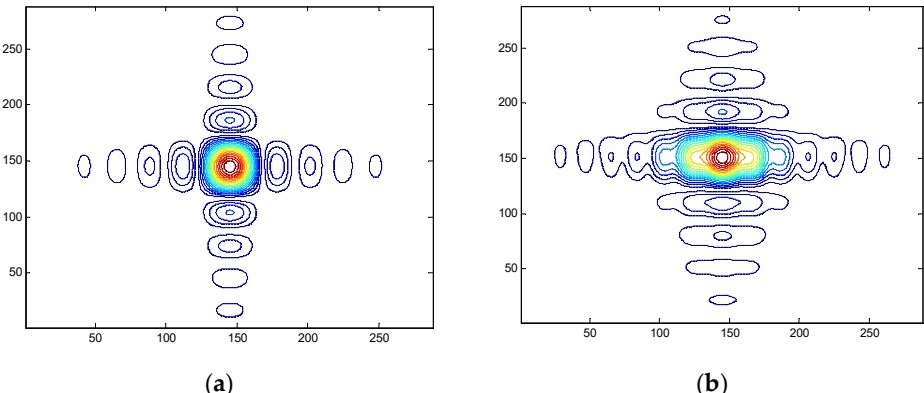

| (a) | (b) |

**Figure 11.** Imaging results of traditional approach: (**a**) center point; (**b**) edge point.

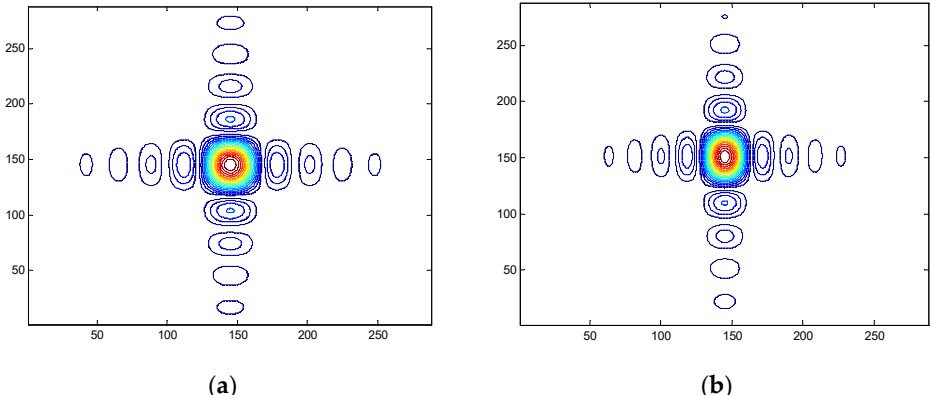

| (a) | (b) |

**Figure 12.** Imaging results of the proposed approach: (**a**) center point; (**b**) edge point.

**Table 2.** Azimuth imaging quality with motion error.

|  | Proposed Algorithm | | | Traditional Algorithm | | |
|---|---|---|---|---|---|---|
|  | IRW | PSLR | ISLR | IRW | PSLR | ISLR |
| Center Point | 0.20 | −13.84 | −10.28 | 0.28 | −13.56 | −10.67 |
| Edge Point | 0.20 | −13.78 | −10.07 | 0.21 | −10.53 | −8.58 |

In comparison, Figure 13a,b show the residual phase error of the traditional approach from [38] and the proposed approach, respectively. It can be seen that under the condition of extremely small incident angle, the residual phase error of the traditional approach is larger than $\pi/4$, which cannot be ignored, while the residual phase error of the proposed approach is notably smaller than $\pi/4$.

Furthermore, Figure 14a,b show the imaging results of the traditional approach from [38] and the proposed approach, respectively. It is evident that the traditional approach fails to achieve satisfactory focusing in Figure 14a due to its non-negligible residual phase error, while the approach proposed in this paper preforms well, which further validates the effectiveness and innovativeness of our work.

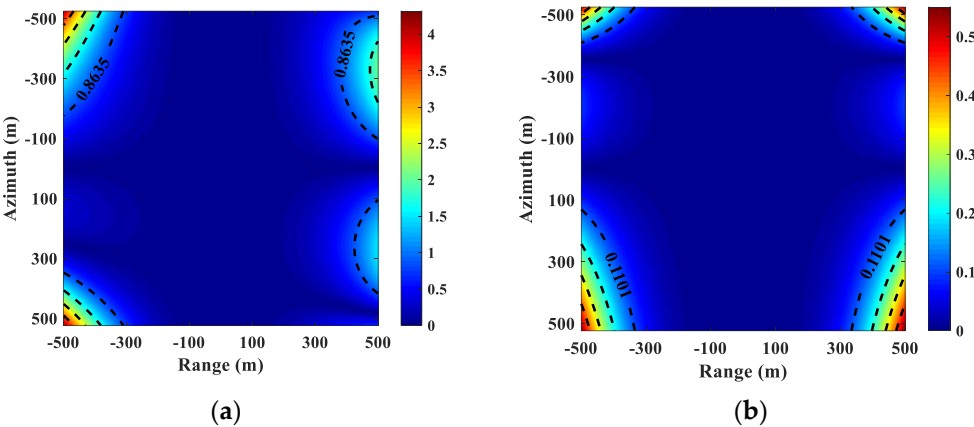

**Figure 13.** Residual phase error of (**a**) the traditional approach; (**b**) the proposed approach.

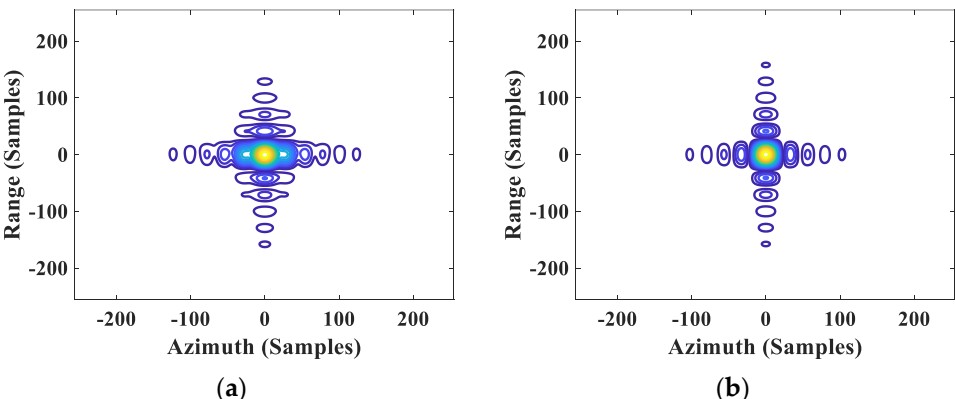

**Figure 14.** Imaging results of (**a**) the traditional approach; (**b**) the proposed approach.

### 4.2. Real Data Experiment Results

The proposed compensation method is highly effective for strip SAR motion compensation and can accurately estimate spatially variant phase error. The performance of the autofocus algorithm, as well as its comparison with the traditional autofocus algorithm [38], was then verified using measured data obtained from a flying experiment using a specific type of radar. The primary system parameters used in the experiment are provided in Table 1.

Figure 15a,b display the imaging results for large scenes processed using the traditional and proposed algorithms, respectively. Furthermore, Figure 16a,b show the imaging results for small scenes, with selected areas of interest magnified.

Figure 16a shows the imaging results processed using the traditional approach, which produces defocused results at the edge of the scene and relatively poor focusing of strong scattering points. These results suggest that spatially variant motion error still have a significant impact on imaging. Figure 16b presents the imaging results obtained using the proposed approach for the spatially variant process, which displays significant improvements in the focusing effect, especially at the edges. The results confirm the effectiveness of the proposed algorithm in compensating for spatially variant motion error and improving the overall imaging quality.

To further evaluate the autofocusing performance of the proposed algorithm, the entropy value was utilized as a quantitative index of image focusing. Generally, the more blurred the image, the greater the uncertainty, and the higher the entropy of the image. The entropy values of the images processed using the traditional approach and the proposed approach were calculated to be 3.6019 and 3.5419, respectively, indicating that the proposed algorithm can effectively correct spatially variant phase error. The enhancement in the

focusing effect, particularly at the edges, confirms that the proposed algorithm is suitable for this scene.

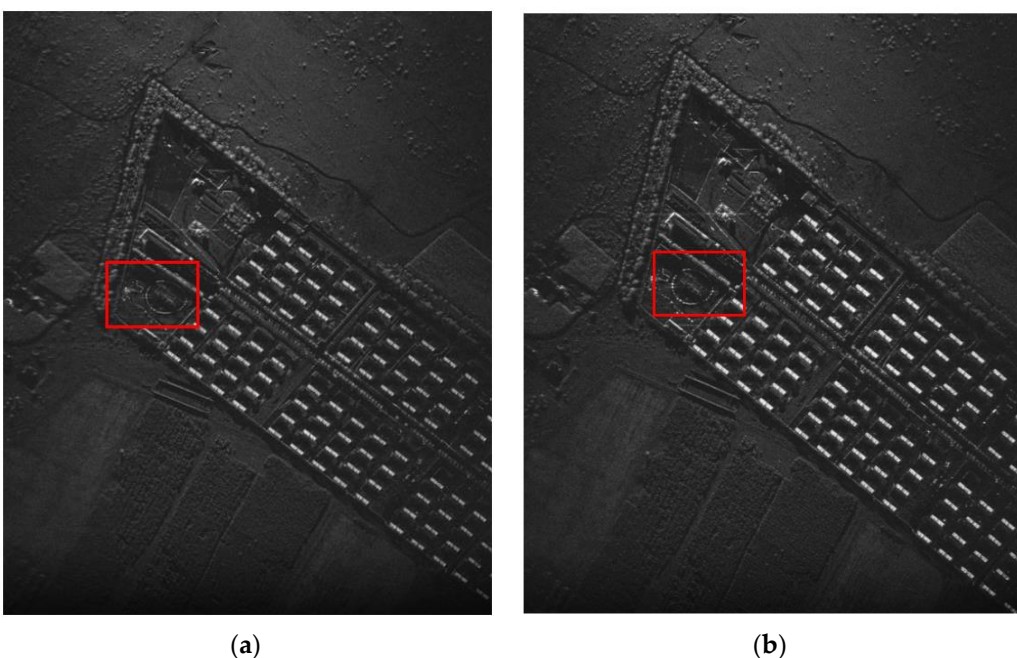

(**a**)　　　　　　　　　　　　　　　　　　　　　　　　(**b**)

**Figure 15.** Real-data imaging results in large scenes: (**a**) traditional algorithm; (**b**) proposed algorithm.

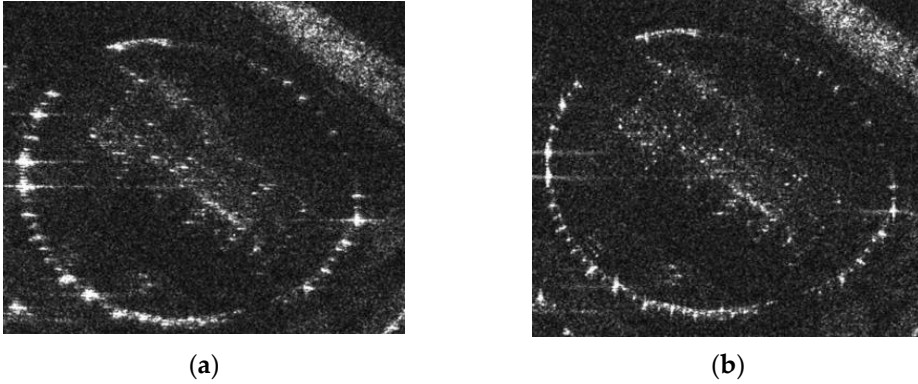

(**a**)　　　　　　　　　　　　　　　　　　　　　　　　(**b**)

**Figure 16.** Real-data imaging results in small scenes: (**a**) traditional algorithm; (**b**) proposed algorithm.

## 5. Conclusions

Motion error, a form of atmospheric interference, poses a significant challenge in the design of remote detection and imaging methods for UAV SAR applications. In this study, we analyzed the practical issues arising from motion error, including envelope error, phase modulations, and NPE, to establish the imaging model of airborne SAR. A statistical threshold was utilized for feature selection, and we developed an improved phase-weighted estimation PGA algorithm that accurately approximates the phase error induced by spatial variation. A composite autofocus approach was developed by combining the improved phase-weighted estimation PGA algorithm with an auxiliary algorithm and building upon the threshold. The auxiliary algorithm compensates for the situation of fewer features present in the scene. The effectiveness and applicability of the approach are verified through both simulation and real data experiments.

**Author Contributions:** Conceptualization, X.Z., S.T. and Y.R.; methodology, X.Z., S.T., Y.R. and J.H.; software, X.Z. and S.T.; validation, S.T., T.J., J.Z., Y.L. and Q.D.; writing—original draft preparation,

X.Z. and S.T.; writing—review and editing, Y.R., C.J. and J.H. All authors have read and agreed to the published version of the manuscript.

**Funding:** This work was supported in part by the National Natural Science Foundation of China under Grant 61971329, Grant 61701393, Grant 62001062, and Grant 61671361; in part by the Natural Science Basis Research Plan in Shaanxi Province of China under Grant 2020ZDLGY02-08; and in part by the Fundamental Research Funds for the Central Universities under Grant ZYTS23153.

**Conflicts of Interest:** The authors declare no conflict of interest.

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
