# Peer review of "Spatially Variant Error Elimination for High-Resolution UAV SAR with Extremely Small Incident Angle"

_remotesensing, doi:10.3390/rs15143700_

Round 1
Reviewer 1 Report
Please see the attached file

Could be improved. See the comments.
Reviewer 2 Report
This manuscript proposes an autofocus algorithm for high resolution UAV SAR with extremely small incident angle. The content is clear and easy to understand. However, the novelty of this paper is not prominent. Therefore, I recommend that the paper could not be published until the following questions are well solved.
1:I suggest that the author include more descriptions of the main challenges posed by high-resolution UAV SAR with an extremely small angle of incidence.
2: There is no need to spend too much space on the conventional PGA and traditional range-variant PGA.
3: The improved phase-weighted estimation PGA in manuscript lacks innovation. Its process is very similar to the EPGA in [1]. Even the notation of their formulas is very similar.
4: The ISLR of imaging results in Table 2 in manuscript are -13.36dB and -13.29dB, deviating from the ISLR of sinc function, -10dB, seriously. However, the impulse response function in Figure 11 is very ideal. That doesn't make sense. I suggest the authors search for and correct the problem.
[1] Zhang Lei. Study on high resolution SAR/ISAR imaging and error Correction[D]. Shaanxi: Xidian University, 2012.
The English expression of the article needs to be improved.
Reviewer 3 Report
The paper presents a method for collecting high resolution SAR from UAVs which can compensate for the variations of the mobile radio platform's motion from the ideal straight paths created by wind or other small errors likely to be created errors in the navigation system.
My main comments on the paper are below:
1. The geometric model in Section 2.1 is a little confusing. Figure 1 seems to be showing the trajectory being parallel to the x-axis but the equations and text seem to indicate that the motion is parallel to the y-axis. An angle \beta is referred to in the text. It would be easier if this angle was shown in the diagram.
2. Figure 3 shows regions defined as 'near' and 'far' detections areas but is not clear nor explained why these regions are 'near' and 'far'. There seems to be some explanation missing here.
3. Equation 7 is the linear unbiased minimum variance (LUMV) estimate of the phase error gradient. Is this a novel contribution of this paper? If so, we should have a derivation of this equation. If it is not, then there should be some citation to previous literature where this was derived.
4. The presentation of the PGA algorithm in Section 3.2 is needlessly complex and hard to follow. Variables are introduced without proper definition or being putting into context with previously described concepts. For example, the measurements from which the phase error is being estimated in Equation 11 are not clearly described. New variables such as s_d(j,dot) are defined but not clearly put into context of what measurements the system would have. I would recommend giving this section a proofread and revision.
5. The paper compares the proposed methods with the 'Traditional Method' in Table 2 but what exactly is this 'Traditional Method'? This should be clearly defined. The paper should provide a quantifiable improvement of the new method versus the prior techniques.
I believe that major revisions are required of this paper before I can recommend it for publication.
The English grammar of this paper are fairly good. Proofreads for grammar and spelling are always recommended but I did not have problems understanding this paper due to grammar or other language issues.
Round 2
Reviewer 2 Report
It is clear that the author has put a great deal of effort into revising the manuscript and that some of the problems with the previous version of the manuscript have been corrected as a result of the author's efforts. However, I still think the novelty of this manuscript is not prominent. I have the following questions:
1:The innovation of this manuscript is not clear, whether it is the improved PGA, the statistical threshold selection or the flow chart in Fig. 6 in manuscript? If it is improved PGA, the manuscript is not innovative. If it is the statistical threshold selection or the flow chart, please add more description about the innovation.
2: There is no need to spend too much space on the conventional motion compensation in the introduction.
The English expression of the article needs minor editing.
Reviewer 3 Report
My questions have been answered by the authors in their response to my first review. There are still some minor issues that need to be corrected before publication but given the authors' responses, I don't think another full review will be necessary:
1. The authors corrected Figure 1 but then changed the text to say that the platform is moving along the x-axis which makes the diagram confusion. Please ensure the text and diagram are consistent.
2. The results compare their method with the 'Traditional method' but the text does not identify the 'Traditional Method'. The author's response says that this method is from reference [18]. This reference should be in the paper since the reader's should be able to reproduce this results. I am not sure if a reference from 1991 would give the state-of-the-art methods for imaging. I would recommend comparing your methods with some more recent. I think your method will still give good results but you should be clear that you are comparing your method with recent prior art.
The quality of the English is acceptable.
